# Theory of Order-Disorder Phase Transitions Induced by Fluctuations Based on Network Models

Yonglong Ding

*Beijing Computational Science Research Center, Beijing 100193, China and*
*School of Physics, Beijing Institute of Technology, Beijing 100081, China*
(Dated: January 14, 2025)

Both quantum phase transitions and thermodynamic phase transitions are probably induced by fluctuations, yet the specific mechanism through which fluctuations cause phase transitions remains unclear in existing theories. This paper summarizes different phases into combinations of three types of network structures based on lattice models transformed into network models and the principle of maximum entropy. These three network structures correspond to ordered, boundary, and disordered conditions, respectively. By utilizing the transformation relationships satisfied by these three network structures and classical probability, this work derive the high-order detailed balance relationships satisfied by strongly correlated systems. Using the high-order detailed balance formula, this work obtain the weights of the maximum entropy network structures in general cases. Consequently, I clearly describe the process of ordered-disordered phase transitions based on fluctuations and provide critical exponents and phase transition points. Finally, I verify this theory using the Ising model in different dimensions, the frustration scenario of the triangular lattice antiferromagnetic Ising model, and the expectation of ground-state energy in the two-dimensional Edwards-Anderson model.

Through advancements in computational methods and models tailored for phase transitions, our understanding of these phenomena has become increasingly profound and comprehensive. However, existing algorithms such as Monte Carlo[1–5], tensor networks[6–9], and scaling laws[10–12] all confront significant challenges, namely, a steep increase in computational complexity as the system size grows[13–16]. Consequently, there remains a lack of comprehensive understanding of phase transitions at infinite scales.

Both quantum phase transitions [17] and thermodynamic phase transitions [18] can potentially be induced by fluctuations [19, 20]. However, how fluctuations specifically cause phase transitions remains an area where Landau's theory of phase transitions provides only a phenomenological explanation [21], lacking a clear description of the microscopic mechanisms underlying phase transitions.

Ding[22, 23] recently developed a computational method that directly converts infinite-scale models into network models with finite nodes and constructed a maximum entropy network structure based on the principle of maximum entropy[24]. This approach allows for the estimation of critical points and critical exponents for the Ising model's phase transition. In this paper, I comprehensively enhance this method by proposing a high-order detailed balance relationship based on network models. Taking the maximum entropy network structure as the starting point and building a bridge between fluctuations and phase transitions, this method provides a clear picture of how fluctuations induce phase transitions and offers an estimation method for critical points and critical exponents in ordered-disordered phase transitions under general conditions. I also present frustration scenarios in antiferromagnetic Ising models[25–27] and estimates of

ground-state energy in Edwards-Anderson models[28].

*Theory—* For lattice models, the infinite lattice model is transformed into a network model. Firstly, all possible lattice sites in the lattice model are classified according to the magnitude of their interactions and the spin of the lattice sites themselves. Let $C_{ij}$ represent the weights of different types of lattice sites, where $i$ denotes the spin type of the lattice site itself, and $j$ represents different types of neighbor interactions. Then, different $C_{ij}$ values are treated as different network nodes. If there exists a transformation relationship between different network nodes caused by fluctuations, the two network nodes are connected by a line segment. Finally, the different phases of the lattice model are labeled using the weights of different network nodes. If all lattice sites have spins pointing upwards, the weight of the corresponding network node is set to 1, while the weights of other network nodes are set to 0. This method does not focus on the specific position and momentum of any individual lattice site, but rather on the weights of different types of lattice sites. In other words, it attempts to capture the core physical information by using the weights and changes of different types of lattice sites.

The principle of maximum entropy is a widely applied statistical principle. In this paper, the principle of maximum entropy is used to describe the most random state of a lattice model under certain conditions. Described using the network nodes mentioned earlier, it means that the distribution of weights among different nodes is in its most random state. Knowing the weight of any single node segment for the direct calculation of the transition probabilities of all connected nodes based on factors such as temperature and interaction strength. From this, the weights of all connected nodes can be obtained using the detailed balance equation, and $\sum_{ij} C_{ij} = 1$ the sum of

the weights of all nodes equals 1. The network structure that satisfies the above conditions is referred to as the maximum entropy structure in this paper, denoted as $\mathbb{M}$.

For the case where all spins are identical, the weight of one corresponding network node is 1, while the weights of all other network nodes are 0, and this network node possesses the lowest energy among all network nodes.

For instance, the ground state of the ferromagnetic Ising model, where all spins are oriented in the same direction, falls into this category. In this paper, it is referred to as the single-node structure, denoted as $\mathbb{D}$. Subsequently, the discussion will primarily focus on the phase-transition relationship between single-node structure and the maximum entropy network structure. Other ordered structures are analogous to this situation. Both structures are stable under different conditions.

This paper introduces the concept of intermediate structures $\mathbb{B}$. Specifically, there is no large-scale direct transition between the two structures mentioned above. Instead, the transition process primarily involves converting into an intermediate structure first, and then transitioning from this intermediate structure to another network structure, which is detailed later as passive transformation. Intermediate structures typically serve as the boundary between the maximum entropy network structure and the single-node structure, separating the two stable structures to form a relatively stable state. This boundary does not always exist and may only form after symmetry breaking occurs. $\mathbb{D}$ , $\mathbb{B}$ , and $\mathbb{M}$ can all be represented by different values of $C_{ij}$.

Specifically, in a strongly correlated system, when the spin of a particular lattice site changes, the spin type of that lattice site also changes, resulting in a change in the weight of the network node to which this lattice site belongs. This type of transformation is referred to as an active transformation in this paper. Meanwhile, the network nodes associated with the neighboring lattice sites of this lattice site also undergo changes, and this type of transformation is termed a passive transformation in this context. The active transformation corresponds to a change in $i$ within $C_{ij}$, while the passive transformation corresponds to a change in $j$.

When updating a lattice model by flipping a lattice site, it can be described from two perspectives: actively as $\frac{\partial C_{ij}}{\partial i}$ or passively as $\frac{\partial C_{ij}}{\partial j}$, with both descriptions being equivalent. If each lattice site has $n$ interacting neighboring sites, then the following formula can be obtained.

$$\frac{\partial C_{ij}}{\partial i} = n\frac{\partial C_{ij}}{\partial j} \qquad (1)$$

The probability of active transformation can be directly calculated from the independent variables related to fluctuations. The transition probabilities remain relatively stable before and after the phase transition, making it difficult to directly observe the phase transition.

However, the combination of passive and active transformations can lead to very drastic changes. Therefore, this paper focuses on passive transformation to calculate and describe the phase transition. During the process of passive transformation, both stable structures undergo intermediate structure as boundary before transitioning into the other structure. Nodes closely related to the maximum entropy network structure are referred to as central nodes. The characteristic of central nodes is that the probability of active transformation between different i values is equal to 1. This allows for rapid weight distribution among different $i$ values. Following this, passive transformation leads to changes in different $j$ values, enabling rapid and comprehensive weight distribution through the central nodes. As a result, a significant portion of the weight of a central node reached through passive transformation will be converted to other nodes. Therefore, the weights of central nodes in this paper correspond to the weights of the maximum entropy network structure.

The transition of weights among different nodes induced by fluctuations satisfies the principle of least action, meaning that when a fluctuation occurs, the required energy consumption is determined, and the transition efficiency between different nodes is maximized. This simplifies and directly relates the correspondence between the fluctuation relations of different nodes. The energy consumed by a single-node lattice flip is determined, and the maximum number of boundary lattice nodes that can be transformed into is also directly obtainable.

Let's explore the correspondence of fluctuations in three types of network structures, starting with the flow of lattice site weights among different types of network nodes.

For a lattice site in a single-node network structure, considering only the nearest-neighbor scenario, when this lattice site flips, $n$ lattice sites undergo passive changes. Due to the principle of least action, these $n$ lattice sites transition from the single-node network structure to an intermediate structure. The transition from the intermediate structure to the maximum entropy structure is also accomplished through lattice site flips. Therefore, the transition rule from the intermediate structure to the maximum entropy network structure is to flip one lattice site in the intermediate structure and calculate how many other lattice sites in the intermediate structure can passively transition to central nodes corresponding to the maximum entropy network structure. This determines the transition from lattice sites in the intermediate structure to the maximum entropy network structure. If flipping one lattice site in the intermediate structure cannot result in any passively transitioning lattice sites to the maximum entropy network structure, then multiple lattice sites in the intermediate structure are flipped simultaneously to achieve the effect of passive transition to

the maximum entropy network structure. In the second process, the minimum number of passive flips achieved by flipping a single intermediate lattice site is $k$, also selected based on the principle of least action. Thus, a flip in the single-node network ultimately leads to $[nk]$ lattice sites in the maximum entropy network structure being passively generated, where [] denotes the floor function since the number of particles cannot be fractional. This allows us to obtain the specific correspondence of fluctuations. Specifically, a fluctuation corresponding to a randomly selected lattice site in the single-node structure corresponds to fluctuations in different numbers of lattice sites in the intermediate and maximum entropy structures, with these numbers required to be integers.

The above provides the correspondence between fluctuations, with a minimum of 1 for these fluctuations. This means that the change in the probability of a fluctuation occurring with changes in the independent variable is relatively a minimum of 1. The smallest fluctuation is an active transformation, and different fluctuations are uncorrelated and can be completed independently. That is, the number of lattice sites undergoing fluctuations increases or decreases one by one, while passive transformation fluctuations are derived from active transformation fluctuations.

In this paper, I do not consider the cases of vacancies and double occupations, so fluctuations correspond to flips. The above sections have provided a detailed introduction on how to transition from $\mathbb{D}$ to $\mathbb{M}$ through lattice site flips, with the formula involving $\frac{\partial}{\partial i}$ and $\frac{\partial}{\partial j}$. The following equation can be derived from the principle of least action.

$$\frac{\partial \mathbb{D}}{\partial i} = n\frac{\partial \mathbb{D}}{\partial j} = n\mathbb{B} \tag{2}$$

Here, $\mathbb{B}$ denotes the lattice points that belong to $\mathbb{B}$. The entire formula represents flipping a lattice point that belongs to $\mathbb{D}$, passively generating $n$ lattice points that belong to $\mathbb{B}$. Similarly, due to the principle of least action, $\mathbb{B}$ can be flipped to obtain $\mathbb{M}$, as shown in the following equation: flipping a lattice point that belongs to $\mathbb{B}$ passively generates $k$ lattice points that belong to $\mathbb{M}$.

$$\frac{\partial \mathbb{B}}{\partial i} = k\frac{\partial \mathbb{M}}{\partial j} \tag{3}$$

Therefore, the unit path for transforming $\mathbb{D}$ into $\mathbb{M}$ through flipping can be obtained, as shown in the following equation, where $[nk]$ lattice points belonging to $\mathbb{M}$ will be generated each time, with [] denoting the integer part.

$$\frac{\partial}{\partial i}(\mathbb{D} \Rightarrow \mathbb{M}) = [nk]\mathbb{M} \tag{4}$$

Simplifying the phase transition caused by fluctuations to changes in the weights of three network structures,

and more specifically, to the flow of weights among these three different network structures. There are multiple scenarios for the transition relationships among the three network structures, but due to the principle of least action, the corresponding relationship for such transitions becomes unique. Different network structures have different step sizes in the changes caused by fluctuations. If the number of fluctuations occurring per unit time in the single-node structure is continuous during the continuous change of the independent variable, then based on the correspondence relationship between fluctuations mentioned above and the correspondence relationship between central nodes and the maximum entropy structure, the passive fluctuation step size of central nodes caused by fluctuations can be seen as $[nk]$. In other words, the change in the weight of central nodes due to passive changes can be regarded as having a step size of $[nk]$. That is, during a complete weight transfer process from the single-node structure to the maximum entropy structure, the smallest unit of increase in the weight of central nodes is $[nk]$, meaning that at least $[nk]$ lattice sites are simultaneously generated through passive transitions. Therefore, among all transitions, the weight-related changes associated with the maximum entropy structure account for a weight of $P$ among all changes with a step size of $[nk]$.

Next, this work directly calculate the probability of the transformation occurring. When actively flipping a lattice point belonging to $\mathbb{D}$, the passively transformed lattice point may belong to either $\mathbb{D}$ or $\mathbb{B}$. This work decompose the transformation from $\mathbb{D}$ to $\mathbb{M}$ into two parts: $\mathbb{D}$ to $\mathbb{B}$ and $\mathbb{B}$ to $\mathbb{M}$.

$$\frac{\partial}{\partial i}(\mathbb{D} \Rightarrow \mathbb{M}) = \frac{\partial}{\partial i}(\mathbb{D} \Rightarrow \mathbb{B}) \oplus \frac{\partial}{\partial i}(\mathbb{B} \Rightarrow \mathbb{M}) \tag{5}$$

In this context, $p$ represents the probability of transitioning to the maximum entropy structure in the absence of correlations, which can be given directly. Since the step size of the transition is $[nk]$, this work only consider the situation where $[nk]$ lattice sites change simultaneously. This is equivalent to randomly selecting $[nk]$ lattice sites, and The probability associated with transitioning to the maximum entropy structure is the likelihood of being related to it, excluding the scenario where all $[nk]$ sites are unrelated to the maximum entropy structure. sites are unrelated to the maximum entropy structure, i.e., flipping one node results in $n$ passive transitions. If these $n$ lattice sites are all unrelated to the maximum entropy structure, then the selected $n$ lattice sites belong to the single node structure or the intermediate structure. From this, the follow formula can be derived.

The probability of a complete transformation process that is unrelated to $\mathbb{M}$ is

$$\frac{\partial}{\partial i}(\mathbb{D} \Rightarrow \mathbb{B}) = (1-p)^{[nk]} \tag{6}$$

So the probability of transforming into $\mathbb{M}$ during the entire process, while being unrelated to $\mathbb{D}$, is

$$\frac{\partial}{\partial i}(\mathbb{B} \Rightarrow \mathbb{M}) = 1 - (1-p)^{[nk]} \tag{7}$$

The weight associated with changes involving intermediate structures is necessarily related to the intermediate structures during the transition process, so the probability is 1.

$$\mathcal{T}(\Gamma', \Gamma)P_{eq}(\Gamma) = \mathcal{T}(\Gamma, \Gamma')P_{eq}(\Gamma') \tag{8}$$

Within the system, $\Gamma$ and $\Gamma'$ represent distinct values, with $P_{eq}$ representing their respective weights and $\mathcal{T}$ indicating the transition probabilities between them. From the relationship between active and passive transformations, we can derive the detailed balance relationship satisfied by $\mathbb{B}$ and $\mathbb{M}$ in passive transformations.

$$(\frac{mP_{BM}}{1 - P_{MB}})^n \tag{9}$$

Where $P_{BM}$ represents the transition probability from an intermediate structure to the maximum entropy structure, and $P_{MB}$ represents the probability of transitioning from the maximum entropy structure to an intermediate structure and then returning to the maximum entropy structure.

Where the numerator represents the probability of transitioning into central nodes. If there are $m$ central nodes and the transition probability between central nodes is 1, it will be multiplied by $m$. The denominator represents the probability of transitioning from central nodes to the intermediate structure, which is 1, but there is a certain probability of returning during the transition to the intermediate structure, leading to the form of the denominator.

However, due to the presence of strong correlations, during the transition between the intermediate structure and the maximum entropy structure, situations where only a single lattice site changes do not exist. Only situations where $n$ lattice sites change simultaneously are present. In other words, when only considering the maximum entropy structure and the intermediate structure, one flip corresponds to the mutual conversion of $n$ intermediate structures and maximum entropy structures. There is no situation where a single lattice site is passively converted into an intermediate structure or a maximum entropy structure. The minimum step size of the existing transition is $n$. This means that $n$ lattice sites of the intermediate structure simultaneously convert into the maximum entropy structure, or vice versa.

Therefore, the transition probability of the detailed balance relationship is:

$$1 - (1-p)^{[nk]} = (\frac{mP_{BM}}{1 - P_{MB}})^n \tag{10}$$

This allows us to obtain the complete higher-order detailed balance relationship.

This is a higher-order detailed balance relationship, which differs from the traditional detailed balance relationship. The traditional detailed balance relationship involves the conversion of individual particles, with the minimum unit being 1. However, the presence of strong correlations complicates this situation, as there are at least $n$ particles changing simultaneously in each transition. When calculating the weights, multiple particles changing simultaneously are also considered, and the calculation of weights for different network structures also involves multiple particles changing simultaneously. The process has changed from originally drawing one particle from the sample each time to now drawing $n1$ particles simultaneously from the sample each time, where $n1$ is a constant. Using classical probability, the detailed balance relationship for this situation can be easily obtained.

*Results and discussion*— In Fig 1(a), $C_{ij}$ represents all possible classification and transformation relationships. Different columns signify distinct spins, while different rows indicate varying interaction intensities. Specifically, $C_{15}$ denotes the single-node network structure, $C_{14}$ represents the intermediate structure, and $C_{13}$ and $C_{23}$ represent central nodes. It is readily apparent from the figure that the transformation from the intermediate structure to the maximum entropy structure is $C_{14} \rightarrow C_{13}$, while the transition from the maximum structure to the intermediate structure is $C_{13} \rightarrow C_{14}$ minus $C_{14} \rightarrow C_{22}$. Therefore, specific transformation formulas can be derived. In Fig 1(b), for a particular spin that is identical to the spin of the single-node network structure, the central node, intermediate structure, and single-node network structure of the Ising model in different dimensions can be obtained. For the Ising model in various dimensions, the corresponding fluctuation relationships can be derived and subsequently substituted into the aforementioned higher-order detailed balance relationships to obtain the formulas describing the variation of magnetic induction intensity with temperature for the Ising model in different dimensions.

$$1 - (1-p)^{[nk]} = (\frac{2}{e^{(n-2)/T} - e^{-(n-2)/T}})^n \tag{11}$$

$$\langle m \rangle = (1 - \frac{1}{sinh^n((n-2)/T)})^{1/[nk]}, T \leq T_c \tag{12}$$

$$\langle m \rangle = 0, T \geq T_c \tag{13}$$

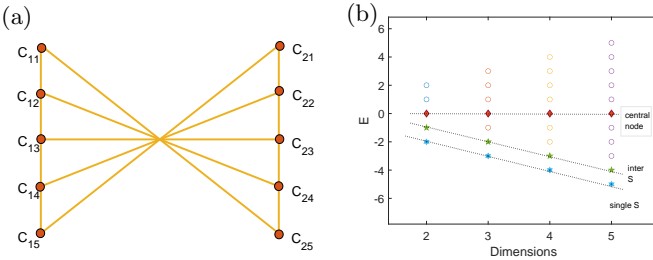

FIG. 1. (a) represents the different nodes and the transformation relationships between them in a two-dimensional Ising model, where different columns signify distinct spins, and different rows from bottom to top indicate increasing interaction strengths. (b) represents, for a specific spin, the single-node structure, intermediate structure, and central node in Ising models of different dimensions, denoted respectively by a rhombus, a pentagram, and a hexagon.

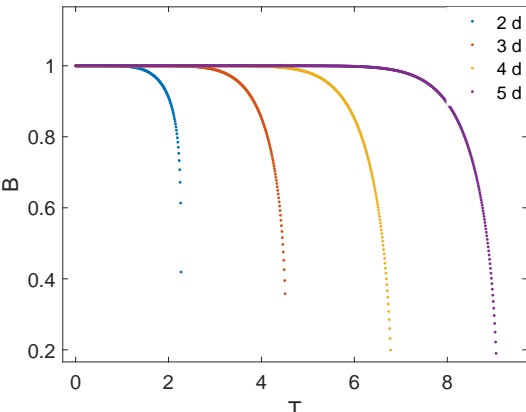

FIG. 2. The lines of different colors in the graph respectively represent the variation of magnetic induction intensity with temperature in two-dimensional, three-dimensional, four-dimensional, and five-dimensional Ising models.

As shown in Fig 2, by substituting the fluctuation and transformation relationships of the Ising models from two to five dimensions, we can obtain the critical exponents as $1/8$, $1/3$, $1/2$, $1/2$, respectively, along with the phase transition points.

*Conclusion* Addressing the issue that existing theories do not clearly describe how fluctuations lead to order-disorder phase transitions, this paper employs a method of networking lattice models and the principle of maximum entropy to classify phases under different independent variables into single-node network structures, maximum entropy network structures, and intermediate structures, which correspond to ordered phases, disordered phases, and the boundaries between the two phases, respectively. The process of phase transitions is mapped onto changes in the weights of these three network structures, and the flow of weights between network nodes clearly delineates this process.

The traditional detailed balance equation may not necessarily apply in strongly correlated systems because transitions between states for particles are not completed one by one but rather by multiple particles simultaneously. Furthermore, there exists a correspondence between fluctuations occurring in different phases, naturally requiring that the slowest fluctuation has a step size of 1, so that fluctuations in other phases, derived from this correspondence, are integer multiples of the slowest fluctuation. Based on these conclusions and classical probability, a high-order detailed balance relationship similar to continuity equations is derived.

Finally, using the high-order detailed balance relationship, the critical exponents and phase transition points of the Ising model in different dimensions are given, and the critical exponents for the 2D to 5D Ising models are specifically calculated as $1/8$, $1/3$, $1/2$, and $1/2$, respectively. Furthermore, the rationality of the theory presented in this paper is further verified through two additional problems: the antiferromagnetic frustration scenario on a triangular lattice and the estimation of the ground state energy of the two-dimensional Edwards-Anderson model.

*Acknowledgments*— This paper is supported by the National Natural Science Foundation of China-China Academy of Engineering Physics(CAEP)Joint Fund NSAF(No. U2230402).

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

# Supplemental Material for
# "Theory of Order-Disorder Phase Transitions Induced by Fluctuations Based on Network Models"

### A: Frustration in Antiferromagnetic Ising Model

Firstly, the antiferromagnetic model on a triangular lattice is transformed into a network model. For simplicity, the initial consideration is limited to the cases where each lattice site has only two possible spin orientations: up and down. For lattice sites with spins up, similar to the approach in the paper, classification is based on the number of neighboring sites with different spin orientations, regardless of their sequence. Among these, the lattice site with the lowest energy is the one surrounded by six neighboring sites all with spins down. This allows all possible lattice sites with spins up to be categorized into 7 classes, corresponding to 7 network nodes. Similarly, for lattice sites with spins down, they can also be converted into 7 network nodes. During the Monte Carlo updating process, all types of lattice sites that can undergo direct transitions are connected by lines. Thus, the entire antiferromagnetic triangular lattice model is transformed into a model with 14 network nodes.

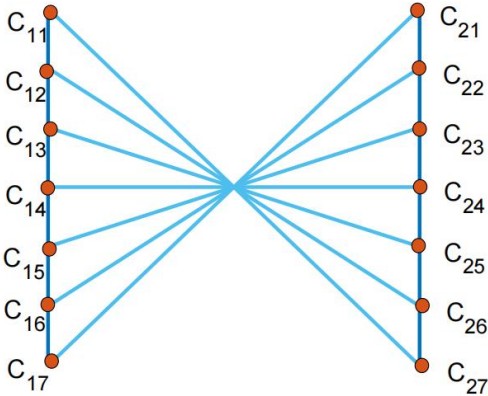

FIG. SM-1. Different network nodes represent different types of spin lattice points, while the line segments represent all possible transformation relationships.

Next, I'll start with a simple scenario, assuming that the spins of different lattice sites can only be in two states: spin-up and spin-down. In the case of antiferromagnetism, for a triangular lattice, the lowest energy configuration occurs when two lattice sites have the same spin and the third site has the opposite spin. It's straightforward to observe that the lowest energy state arises when all triangles adhere to this configuration. Does such a state exist? The answer is yes. This state emerges when one row of lattice sites has spins up, followed by another row with spins down, alternating in this pattern. In the corresponding network model, this corresponds to nodes $C_{15}$ and $C_{25}$ both having weights of 0.5, while other nodes have weights of 0. At extremely low temperatures, this state is stable because flipping any lattice site during Monte Carlo updates would require energy. Therefore, this state is stable.

However, at higher temperatures, the system should conform to the maximum entropy model. The rods representing antiferromagnetic frustration in a triangular lattice are directly related, meaning that the three rods forming a triangle cannot simultaneously be negative. This situation differs from the Ising model in different dimensions, so classical probability methods cannot be directly used to calculate the weights of different nodes at various temperatures. Instead, we can classify all possible types of triangular lattice sites. At different temperatures, different types of triangles have different energies and thus different weights. Subsequently, we can use probability formulas to calculate the weights of different types of lattice sites at various temperatures. The blue rods indicate spins that are the same, while the yellow rods indicate spins that are opposite (note: originally, both blue and yellow rods were described as indicating the same spin, which is corrected here to reflect their distinct meanings in the context of antiferromagnetism).

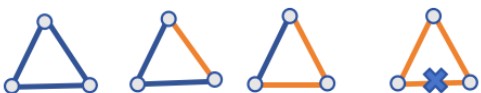

FIG. SM-2. The blue lines represent ferromagnetic interactions, while the yellow lines represent antiferromagnetic interactions. The fourth scenario depicted in the diagram does not exist.

Through the above analyses, we have obtained the scenarios at extremely low temperatures and at high temperatures. It is evident that a transition from the extremely low temperature scenario to the high temperature scenario is possible. Conversely, can the high temperature scenario transition to the extremely low temperature scenario? The answer is yes. During the process of lowering the temperature, the weights of nodes with lower energy increase continuously. However, $C_{16}$ and $C_{26}$ are obviously unstable because they are inevitably surrounded by other lattice sites. In contrast, $C_{15}$ and $C_{25}$ are stable. Therefore, when the temperature drops to a certain level, the probability of transition from $C_{16}$ and $C_{26}$ to other nodes still exists. But since $C_{15}$ and $C_{25}$ are very stable, the transition from $C_{15}$ and $C_{25}$ to $C_{16}$ and $C_{26}$ is unlikely to occur. Consequently, the weights gradually concentrate on $C_{15}$ and $C_{25}$, forming the first scenario. The situations for $C_{17}$ and $C_{27}$ are similar.

Next, let's explore the transition rule from order to disorder. The central points, which are also the centers of maximum entropy, are obviously $C_{14}$ and $C_{24}$. The transition of these nodes does not result in a change in energy, and the close relationship between these nodes and the maximum entropy network is discussed in detail in the article. Since $C_{14}$ and $C_{24}$ are directly connected to $C_{15}$ and $C_{25}$, there are no boundary nodes. This situation implies that there is no phase transition point between the two phases. Assuming each bond has a value of 1, and in this problem $n$ is 6, the formula for the weight variation with temperature in the maximum entropy network structure $p$ is given as follows.

$$p = \left(\frac{2}{e^{(n-2)/T} - e^{-(n-2)/T}}\right)^n (1 - p) = \left(\frac{2}{e^{4/T} - e^{-4/T}}\right)^6 (1 - p) \tag{SM-1}$$

### B: Expectation of Ground State Energy in the Two-Dimensional Edwards-Anderson Model

The relationship between frustration and ground state energy has been established by the algorithm mentioned above. Next, we will use the number of frustrations to derive the ground state energy.

If the probability of $J = \pm 1$ taking a positive value is $q$, then for the two-dimensional Edwards-Anderson model on a square lattice, the expression for all possible combinations of $J$ values across the lattice can be written as

$$(q + (1 - q))^4 = q^4(1 - q)^0 + 4q^3(1 - q)^1 + 6q^2(1 - q)^2 + 4q^1(1 - q)^3 + q^0(1 - q)^4 \tag{SM-2}$$

, where the expansion represents the sum of probabilities for all configurations of J

Firstly, let's consider the expectation of the ground state energy in the two-dimensional Edwards-Anderson model. In this case, the weights for both positive and negative $J$ are 0.5. This allows us to calculate the weights of different types of lattice sites.

Each square lattice has four bonds, and there are five types of bonds, with equal weights of 0.5 for both positive and negative $J$. From this, we can deduce the weights of different square lattices.

The condition for frustration to form is when the number of negative $J$ is either 1 or 3. Only in these cases will a frustrated lattice be created. Each frustrated lattice corresponds to a lattice with the lowest energy. Therefore, there must be a bond with a value of positive 1, and this bond can be shared by two lattices. From this, we can infer the ground state energy of the entire model.

The probability $P$ for a frustrated lattice is

$$P = 4q^3(1 - q)^1 + 4q^1(1 - q)^3 = 0.5 \tag{SM-3}$$

Furthermore, the lowest ground state energy can be estimated by the number of frustrated lattices is $-1 \times 0.5 - 2 \times 0.5 = -1.5$.

Therefore, the weight $p$ of the maximum entropy network structure can be obtained as a function of temperature.

$$1 - (1 - (p - 0.5z))^8 = \left(\frac{2}{e^{2/T} - e^{-2/T}}\right)^4, p - 0.5z \geq 0 \tag{SM-4}$$

$$z = \left(\frac{2}{e^{(n-2)/T} - e^{-(n-2)/T}}\right)^n (1-p) = \left(\frac{2}{e^{2/T} - e^{-2/T}}\right)^4 (1-z) \tag{SM-5}$$

Subtracting $0.5z$ from p is because at extremely low temperatures, the probability associated with the maximum entropy structure can also be transformed through the boundary structure, with a related probability of $0.5z$, without needing to start the transformation from a single-node structure. The relationship between the boundary structure and the maximum entropy structure satisfies Eq SM-1, thus leading to the above formula.