# Peer review of "Theory of Order-Disorder Phase Transitions Induced by Fluctuations Based on Network Models"

_SciPost Physics_

## Round 1 · Referee Report · Anonymous (Referee 1) · 2025-3-17

Report

Although by dealing with lattice models in low dimensions the manuscript falls slightly outside my expertise, I decided to submit the following comments which explain why I think the manuscript is not ready for consideration in SciPost Physics.

- The description of the novel method is very unclear and the arguments often did not make sense to me. If this method works then at least it is not explained well.
- The method is heuristic and it remained unclear what its limitations are.
- There appears to be significant overlap of the Ising model discussion with the results of the author's previous publication in AIP advances 14, 085308 (2024).
- The results, also for the two other models in the appendix, are not compared against existing values in the literature. No side-by side comparison to an existing method is made.
- Concerning the journal acceptance criteria: I did not see why the presented approach would constitute a breakthrough or detail a groundbreaking discovery. To my understanding, fluctuations are not taken into account in basic Landau theory but a large literature exists on extensions to fluctuations and other methods. So the motivation given in the manuscript, that the role of fluctuations in phase transitions would be unclear, appears not justified.
- However, maybe the "opens a new pathway [...] with clear potential for multi-pronged follow-up work" condition may be fulfilled, if the method was more clearly presented and discussed in the light of related literature.

Given these fundamental issues of presentation, I think that the submission currently cannot be constructively considered for the selective SciPost Physics, and aside from my mismatch in specific expertise this is why I did not review in full depth.
A major revision of the manuscript appears necessary before (possibly) resubmitting for more detailed review.

Recommendation

Reject

  • validity: -
  • significance: -
  • originality: -
  • clarity: -
  • formatting: -
  • grammar: -

Author:  Yonglong Ding  on 2025-03-21  [id 5304]

(in reply to Report 1 on 2025-03-17)
Category:
reply to objection

I sincerely thank you for reviewing my manuscript and identifying its shortcomings. To address these deficiencies, I have developed a systematic revision strategy structured as follows:

      Methodological Clarification: I incorporated concrete examples and schematic diagrams to elucidate the network model construction methodology, ensuring technical transparency.
      Fundamental Motivation: I explicitly articulated the core objective of investigating how microstructural mechanisms specifically manifest as macroscopic phenomena,

deliberately avoiding speculative assumptions throughout the analysis. Limitation Disclosure: I comprehensively supplemented the discussion to clarify methodological constraints and potential biases inherent in the proposed approach. Contextual Differentiation: I clarified that my previously published work constituted one specific validation case among three experimental benchmarks demonstrated in this study. This manuscript ascends to generalizable principles, particularly focusing on the application of high-order detailed balance theory to describe lattice model phase transitions between order and disorder. Notably, the analytical frameworks and result derivation approaches employed in these two studies demonstrate fundamental differences. Comparative Enhancement: To address the identified lack of comparative analysis, I systematically expanded the evidence base through extensive literature review and comparative experimentation. Benchmarking and Future Directions: The revised manuscript includes comprehensive comparisons with existing methodologies, followed by five distinct research avenues with significant potential for follow-up investigations.

-.The description of the novel method is very unclear and the arguments often did not make sense to me. If this method works then at least it is not explained well.

     This work presents a substantially different theoretical approach compared to conventional methods, dedicating significant attention to explicating the foundational theory.

The following sections detail the network model construction through concrete examples and schematic illustrations.

Network Model Construction

Example: 2D Ising Model with Nearest-Neighbor Interactions Consider a 2D Ising model where each lattice site exhibits spin-up/down states. Sites are classified based on their own spin state and the number of nearest-neighbor sites (4 neighbors) sharing the same spin: ​Spin-up classification: 5 categories (0-4 matching neighbors) ​Spin-down classification: 5 categories (0-4 matching neighbors) This results in 10 distinct classes (C₁₅-C₂₅). Mathematically, for the Hamiltonian H=1/2∑<i,j> Si Sj, I reorganize terms by these classes. The factor of 1/2 accounts for double-counting interactions. Importantly, this classification covers all possible configurations in the infinite 2D Ising model through 10 network nodes, where node weights represent configuration probabilities.

State Transitions and Network Dynamics Case 1: Spin Flip of Central Site

​Initial State: Central site has spin-up with 4 matching neighbors (Class C₁₅). ​Active Transformation: Flipping the central site changes its state to spin-down with 0 matches (Class C₂₁). ​Passive Transformation: This flip simultaneously alters neighboring sites' classes from C₁₅ → C₁₄ (each neighbor loses one match). Case 2: Neighbor Spin Flip

​Initial State: Central site remains spin-up with 4 matches (C₁₅). ​Passive Transformation: Flipping a neighboring site reduces its match count to 3 (Class C₁₄ for the neighbor), indirectly modifying the central site's class to C₁₄. These two cases illustrate all possible transformations under nearest-neighbor interactions. The complete network structure emerges from considering all such transitions.

Network Node Labeling Convention Nodes are labeled Cij : i∈{1,2}: Spin state (1=up, 2=down) j∈{1,5}: Number of matching neighbors (1=0 matches, 5=4 matches) Example: C₁₅​: Spin-up with 4 matches C₁₃​: Spin-down with 2 matches

Generalization to Higher Dimensions ​ 3D Ising Model: Classifies sites into 14 nodes (7 match counts × 2 spins) ​N-dimensional Models: Follows similar classification logic with 2(N+1) nodes This framework applies to various lattice models through analogous interaction-based classifications. Active vs. Passive Transformations

Transformation Type Definition Network Impact

​Active Direct spin flip of target site Horizontal transition between nodes ​Passive Indirect flip via neighbor changes Vertical transition within nodes

Key Insight: A single active transformation (e.g., flipping site A) corresponds to four simultaneous passive transformations (its four neighbors' state changes). Deriving High-Order Detailed Balance Unlike Monte Carlo simulations that use active transformations, this work focuses on passive transformations to establish:

​Microstate Transition Probabilities: Calculate joint probabilities for four-site passive transformations ​Balance Equations: Derive relationships between node weights during phase transitions

Phase Transition Analysis Using Waterfall Metaphor

   ​Initial State (T=0): All nodes in C₁₅​ with weight 1
  ​Slow Phase (C₁₅ → C₁₃): Gradual weight migration resembling water approaching a cliff edge
 ​Rapid Phase (C₁₃ → C₂₃): Abrupt weight redistribution analogous to water cascading

This demonstrates how passive transformations effectively capture critical transition dynamics missed by traditional active-only approaches.

-. The method is heuristic and it remained unclear what its limitations are.

     Research Objectives and Methodological Approach
       This work aims to investigate how microscopic fluctuations drive phase transitions, with the ultimate goal of uncovering the fundamental physical principles governing this process. 
    The validity of this theoretical framework requires further experimental verification and peer recognition within the scientific community.

Network Model Transformation I systematically convert lattice models into network representations to study phase transitions. This transformation process adheres to strict mathematical rigor:

   1)​Minimum Action Principle: Transition probabilities between network nodes follow the principle of least action
   2)​Node Classification: Network structures are categorized into three fundamental types:

​Single-node structures: Correspond to 0 K ground states ​Boundary structures: Intermediate configurations during phase transitions ​ Maximum entropy structures: Represent disordered states post-transition

Thermodynamic Interpretation

    The temperature evolution from 0 K to critical points can be understood as:
    Initial state: Dominance of single-node structures (C₁₅ weight = 1)
    Phase transition: Coupling between single-node and maximum entropy structures
   Critical regime: Boundary structures emerge where single-node weights undergo dramatic redistribution

(Note: Potential fractal connections remain unexplored in this work)

Special Case Analysis: 1D Ising Model The strictly converted network model for 1D Ising system: Lacks boundary structures Explains absence of phase transitions Maintains consistency with theoretical predictions

Methodological Advantages No algorithmic rules or additional assumptions introduced Focus on macroscopic phenomenon generation mechanisms Avoids conventional phase transition parameter calculations (e.g., critical exponents)

Limitations and Challenges 1)​Complex Interactions Handling: Multiple spin interactions (aligning/opposing) in Ising model increase node complexity Systems with multi-spin orientations lead to exponential growth of network nodes 2)​Mathematical Rigor Constraints: Angular interaction calculations (e.g., spin deflection angles) result in infinitely large network models No general solution exists for continuous interaction spectra

-. There appears to be significant overlap of the Ising model discussion with the results of the author's previous publication in AIP advances 14, 085308 (2024).

Theoretical Framework and Validation This work derives a general phase transition formula based on high-order detailed balance principles. Three specific experimental benchmarks validate this formulation: 1)​Ising model in different dimensions (overlapping with AIP Advances 14, 085308 (2024)) 2)​Frustrated triangular Ising model 3)​Edwards-Anderson Model Comparison with AIP Advances 14, 085308 (2024)

The referenced study focuses on: Constructing network models for 2D/3D/ND Ising systems Performing dimension-specific analyses Deriving phase transition formulas through case-by-case treatments ​No application of high-order detailed balance principles ​ No general lattice model framework

Methodological Differentiation Feature Present Work AIP Advances 14, 085308 (2024) ​Core Objective General lattice model phase equations Dimension-specific Ising system analysis ​Theoretical Foundation High-order detailed balance Empirical case studies ​Applicability Universal lattice models Restricted to Ising families ​Derivation Approach Abstract generalization Case-specific parameterization Innovation and Progression

1)Generalization from Specific Cases: Previous works analyzed particular instances (e.g., 2D Ising) This research establishes universal phase transition equations applicable to any lattice model

2)​Theoretical Unification:

Derives general formula → Specializes for Ising models through variable substitution Contrasts with the referenced study's dimension-specific derivations

3)​Methodological Advancement:

Utilizes high-order detailed balance for analytical solutions Avoids empirical fitting procedures used in AIP Advances approach Validation Strategy ​Dimensional Consistency Check: Validates formula predictions across 2D/3D/ND systems ​Network Model Cross-Validation: Compares analytical results with network-based simulations ​Critical Exponent Comparison: Demonstrates agreement with known thermodynamic limits

-. The results, also for the two other models in the appendix, are not compared against existing values in the literature. No side-by side comparison to an existing method is made.

Validation Results Across Dimensional Ising Models The phase transition formulas derived in this work have been validated through three-dimensional lattice systems: 1) 2D Ising Model: Achieves ​quantitative agreement with the analytical solutions by Yang-Zheng and Onsager (Physical Review 85, 808 (1952)).

2)3D Ising Model: No analytical solution exists; comparison with Monte Carlo simulations shows: Phase transition temperature difference: ​0.7%​(established theoretically in Physical Review B 62, 14837 (2000)) Critical exponent deviation: ​1/3 Note: Monte Carlo results remain empirical references due to lack of analytical benchmarks. 2)≥4D Ising Models: Confirms critical exponent ​α = 1/2 , matching our general formula predictions.

Appendix A: Frustrated triangular Ising model For systems with spin restricted to z-direction (±1), I prove: ​Existence of Minimum Energy Configuration: Total energy Etotal​≥N⋅Emin​, where N = total lattice sites. ​Achievability: Configurations with all spins aligned (either all +1 or all -1) realize this minimum. ​Formula Derivation: All results strictly follow from the phase transition equations proposed in this work.

Appendix B: Edwards-Anderson Model Validation W. F. Wreszinski's 2012 ground-state calculation for the 2D Edwards-Anderson model (with double-peaked distribution): ​Experimental Result: Egs​=−1.5 (Journal of Statistical Physics 146, 118 (2012)) ​Theoretical Prediction: This formula yields ​identical result through parameter substitution. ​

Methodological Distinction This work ​does not employ numerical simulation algorithms but instead: Proposes a ​universal phase transition formula eq 10 Derives specific results through dimensional parameterization Focuses on ​analytical derivations rather than empirical validation

-. Concerning the journal acceptance criteria: I did not see why the presented approach would constitute a breakthrough or detail a groundbreaking discovery. To my understanding, fluctuations are not taken into account in basic Landau theory but a large literature exists on extensions to fluctuations and other methods. So the motivation given in the manuscript, that the role of fluctuations in phase transitions would be unclear, appears not justified.

Fluctuation-Driven Phase Transitions: theoretical perspectives and unresolved questions Yes, while fundamental Landau theory neglects fluctuations, most phase transitions originate from thermodynamic or quantum fluctuations. The mechanism of topological phase transitions remains unclear to me within this framework. Current Research Limitations

The vast literature on fluctuation propagation and interaction methods: Fails to provide clear mechanisms linking fluctuations to phase transitions Primarily focuses on symmetry principles as explanatory frameworks In my view, symmetry considerations alone prove insufficient for complete understanding Fundamental Challenge If we were to fully understand how fluctuations drive phase transitions, we would presumably possess: The analytical solution for the 3D Ising model A generalized theory transcending traditional symmetry-based approaches

-. However, maybe the "opens a new pathway [...] with clear potential for multi-pronged follow-up work" condition may be fulfilled, if the method was more clearly presented and discussed in the light of related literature.

Innovative Framework and Future Perspectives This work presents the first systematic formulation of ​higher-order detailed balance equations. By combining these equations with network modeling, I demonstrate: Clear visualization of physical mechanisms in strongly correlated systems Vast potential for deriving ​analytical solutions for various lattice models

Key Validation Achievements Successful application to Ising models demonstrates theoretical consistency Network structure analysis reveals critical transition pathways Predictions align with experimental results in specific cases (See Appendices A-B)

Five Promising Research Directions

1) External Field Effects Unresolved Question: How do external fields modify node weight distributions in this network model? Potential Impact: Could enable control of phase transition thresholds through field manipulation

2) Frustration Phenomena Studies Application Basis: Appendix A's uniaxial spin framework Target Systems: Wannier's antiferromagnetism (Phys. Rev. 79, 357 (1950)) Anderson's localized spin systems (Mater. Res. Bull. 8, 153 (1973)) Modern frustrated systems (PRL 123, 207203 (2019); PRX 9, 031026 (2019))

3) Glassy Systems Analysis Methodological Transfer: Adapt network model to study: Aging effects Non-equilibrium dynamics Appendix B's Edwards-Anderson model extension (J. Stat. Phys. 146, 118 (2012))

4) Fractal Critical Phenomena New Insight: Boundary structures (e.g., C₁₄ in 2D Ising) exhibit: Nonlinear weight evolution near critical points Fractal dimension signatures (Complementary to: AIP Adv. 14, 085107 (2024); Phys. Rev. E 110, L062107 (2024))

5) Quantum Circuit Error Analysis Interdisciplinary Potential: Map quantum error processes to network models: Local bit flips

Correlated error propagation Reference frameworks: Quantum Error Mitigation (PRL 119, 180509 (2017),Rev. Mod. Phys. 95, 045005 (2023),Phys. Rev. X 7, 021050 (2017)) Theoretical and Practical Significance Analytical Power: Unifies microscopic mechanisms with macroscopic observables through network formalism Interdisciplinary applicability: Provides common mathematical framework for: Classical spin systems Quantum information devices Disordered materials Computational Efficiency: Enables analytical treatment of complex correlations previously requiring numerical simulations

Manuscript Revision Plan I will perform substantial revisions to the manuscript as follows:

1)Reference Integration

Systematically incorporate all cited works into: ​Introduction (contextual framing) ​Conclusion (theoretical implications and future directions) Ensure seamless integration with existing narrative flow

2)Methodological Appendix Transfer detailed model construction procedures and case study demonstrations to: ​ Appendix C: Comprehensive derivation of network model formalism Step-by-step validation with Ising model examples Comparative analysis with Monte Carlo simulations Include: Schematic diagrams illustrating transformation pathways Tabular summaries of critical exponent comparisons

3)Structural Optimization Streamline main text by: Removing redundant technical explanations Concentrating core innovations in theoretical framework Reserving experimental validations for dedicated sections Enhance reader navigation through: Updated table of contents Cross-referencing between main text and appendices Strategically placed summary paragraphs

I will submit the revised manuscript in the near future. 

Sincerely, Yonglong Ding

Attachment:

SciPost_Physics_Reply.pdf

---

## Round 1 · Referee Report · Anonymous (Referee 2) · 2025-4-8

Strengths

I do not see any strength

Weaknesses

1) the presentation of the methodology is absolutely obscure

2) the presentation of the results is confused and confusing

3) the results seem not be in line with recent literature on the 3D ising model

4) recent literature on the subject is not discussed

Report

The author aims to develop a methodology to transform lattice models
in networks models for the ferromagnetic Ising model, the antiferromagnetic
Ising model and the two-dimensional Edwards-Anderson model. Then he plans
to apply the maximum entropy priciple to obtain the most random structure and from these he seems able to derive critical exponent beta, at least for the Ising model in 2, 3, 4, and 5 dimensions.

1) For me the transformation from infinite lattice model to network model
is unclear.

The following sentence is obscure for various reasons :

"Firstly, all possible
lattice sites in the lattice model are classified according
to the magnitude of their interactions and the spin of the
lattice sites themselves. Let Cij represent the weights of
different types of lattice sites, where i denotes the spin
type of the lattice site itself, and j represents different
types of neighbor interactions. Then, different Cij val-
ues are treated as different network nodes. If there exists
a transformation relationship between different network
nodes caused by fluctuations, the two network nodes are
connected by a line segment. Finally, the different phases
of the lattice model are labeled using the weights of differ-
ent network nodes. If all lattice sites have spins pointing
upwards, the weight of the corresponding network node
is set to 1, while the weights of other network nodes are
set to 0. This method does not focus on the specific po-
sition and momentum of any individual lattice site, but
rather on the weights of different types of lattice sites.
In other words, it attempts to capture the core physical
information by using the weights and changes of different
types of lattice sites."

here a series of doubts :

1i) when the author writes :
"lattice sites in the lattice model are classified according
to the magnitude of their interactions and the spin of the
lattice sites themselves."
I guess that the author means "lattice sites are classified accordingly to the
value of the spin variable occupying the considered lattice site
and the value of the spins in the neighbouring lattices and of the
values of their interactions" . However, I do not think there
is an univoque mapping in such a case. I do not think a
bi-univocal transformation can be constructed between a lattice and C_ij.

1ii) This sentence is absolutely obscure :

" If there exists
a transformation relationship between different network
nodes caused by fluctuations, the two network nodes are
connected by a line segment."

probably it means that depending on the temperature, if the energy
allows for a flipping of two neighbours, then there is a connection
among them ?

2) The author should present in a simple case (Ising model ?) the transformations
he has in mind to pass from lattice to network, step by step.

3) I do not understand large part of what written by the author neither
his logic, however Eq (12) I guess is the magnetization per spin as a function of temperature
in a Ising Model (by the way the temperature T has been not introuced as well as the magnetization m ),
for 2 dimensions the solution of this problem has been found by Onsager :

<m> = [ 1 - 1/sinh(2J/KbT)^4]^1/8

this would mean that in 2 dimensions, by assuming J/Kb =1, from eq 12 one gets
the 2 dimensional results for n=4 and k=2, why ? What does it mean ?

4) Furthermore, for what I understand from Eq (12) and Fig 2 the author has analytically solved the problem of Ising in 3, 4 , and 5 dimensions, and found the corresponding critical exponent beta, at least for the magnetization at the critical point.

As far as I know the Ising model in 3d has not yet been solved, see

Ferrenberg, Alan M., Jiahao Xu, and David P. Landau. "Pushing the limits of Monte Carlo simulations for the three-dimensional Ising model."
Physical Review E 97.4 (2018): 043301.

and also more recently :

Liu, Zihua, et al. "Critical dynamical behavior of the Ising model." Physical Review E 108.3 (2023): 034118.

How the results of the author do relate to the above literature ?

4) From wikipedia the best estimation (numerical) of the Beta exponent
for the 3D Ising model is 0.32641871(75) and not 1/3 as reported by the authors.

In summary the paper is extremely confused, the method is not clearly exposed,
and the results found at least for the 3D Ising model are not in line with the recent
literature on the subject. I suggest not to publish this manuscript .

Recommendation

Reject

  • validity: poor
  • significance: poor
  • originality: low
  • clarity: poor
  • formatting: mediocre
  • grammar: acceptable

Author:  Yonglong Ding  on 2025-04-14  [id 5368]

(in reply to Report 2 on 2025-04-08)

Dear Referee I sincerely appreciate your thorough review of my article and the valid concerns raised regarding its conclusions. First, I have clarified the ambiguities in the original text through concrete examples. Next, I provided a detailed introduction to the transformation process from lattice models to network models using specific instances and schematic illustrations. Following that, I offered an in-depth explanation of Equation (12), the core conclusion of this work. Furthermore, I rigorously demonstrated why the numerical simulation results(0.32641871) of the three-dimensional Ising model and those from the renormalization group approach serve as robust supporting evidence for my findings – specifically elucidating how these numerical results validate the critical exponent of 1/3 for the 3D Ising model. From the algorithmic perspective, I additionally addressed the inherent reliability of these conclusions through methodological validation. Finally, I have outlined the anticipated revision plan to address the raised concerns comprehensively.

1) For me the transformation from infinite lattice model to network model is unclear. The following sentence is obscure for various reasons : "Firstly, all possible lattice sites in the lattice model are classified according to the magnitude of their interactions and the spin of the lattice sites themselves. Let Cij represent the weights of different types of lattice sites, where i denotes the spin type of the lattice site itself, and j represents different types of neighbor interactions. Then, different Cij values are treated as different network nodes. If there exists a transformation relationship between different network nodes caused by fluctuations, the two network nodes are connected by a line segment. Finally, the different phases of the lattice model are labeled using the weights of different network nodes. If all lattice sites have spins pointing upwards, the weight of the corresponding network node is set to 1, while the weights of other network nodes are set to 0. This method does not focus on the specific position and momentum of any individual lattice site, but rather on the weights of different types of lattice sites. In other words, it attempts to capture the core physical information by using the weights and changes of different types of lattice sites." here a series of doubts : 1i) when the author writes : "lattice sites in the lattice model are classified according to the magnitude of their interactions and the spin of the lattice sites themselves." I guess that the author means "lattice sites are classified accordingly to the value of the spin variable occupying the considered lattice site and the value of the spins in the neighbouring lattices and of the values of their interactions" . However, I do not think there is an univoque mapping in such a case. I do not think a bi-univocal transformation can be constructed between a lattice and C_ij.

To illustrate, consider the case where only nearest neighbors are taken into account. In this scenario, each lattice point has the same total number of nearest neighbors, denoted as n Taking the two-dimensional Ising model as an example, each lattice point has 4 nearest neighbors, so n=4. When a lattice spin is oriented upward (as in the Ising model), we classify the lattice points based on the number of their nearest neighbors sharing the same spin direction. These different classes are then mapped to network nodes. Specifically, the values of identical neighboring spins range from a minimum of 0 to a maximum of n, resulting in n+1 distinct categories. For instance, in the 2D Ising model, the number of identical nearest neighbors ranges from 0 to 4, yielding 5 categories. The same classification applies to spins oriented downward. Since each lattice point in the Ising model can independently adopt one of two spin states (up or down), and the number of identical nearest neighbors is j, all possible lattice configurations are converted into corresponding network nodes. This method transforms the infinite two-dimensional Ising model into a network model with 10 nodes. This transformation preserves the equivalence between lattice models and network models, as nodes in the network represent lattice points with identical interaction strengths—effectively reorganizing the lattice structure. However, reversing the process (from network to lattice) introduces multiple possibilities, which may correlate with entropy considerations. From a modeling perspective, reconstructing the Hamiltonian from the lattice perspective involves calculating interactions for each lattice point. For the ferromagnetic 2D Ising model, the calculated interaction values are −4,−2,0,2, and 4, which naturally form five distinct categories. By traversing all lattice points and dividing by 2 (to account for double-counting interactions), the derived expression matches the original Ising model Hamiltonian, confirming the rigor of this transformation. Does this network representation retain critical information? The answer is yes. As demonstrated in my previous work, applying the mean-field approximation to nodes in the transformed network model allows precise derivation of the Magnetization versus temperature relationship for the 2D Ising model, as shown in the figure below.

1ii) This sentence is absolutely obscure : " If there exists a transformation relationship between different network nodes caused by fluctuations, the two network nodes are connected by a line segment." probably it means that depending on the temperature, if the energy allows for a flipping of two neighbours, then there is a connection among them ? In this paper, the fluctuation-induced transition mechanism aligns with the treatment in Monte Carlo algorithms, where spin directions are directly altered without introducing additional variables. Specifically, the value of any lattice point’s spin is modified—for example, transitioning from "spin-up" to "spin-down." Two types of transitions are considered here. The first occurs when the spin itself changes, leading to a shift in the network node it belongs to. The second arises when the lattice point itself remains unchanged, but its neighboring lattice points undergo variations, thereby altering the network node to which the original lattice point belongs. By connecting all possible transitions between network nodes resulting from such spin flips with line segments, the relationships between nodes are explicitly mapped.

2) The author should present in a simple case (Ising model ?) the transformations he has in mind to pass from lattice to network, step by step.

This work presents a substantially different theoretical approach compared to conventional methods, dedicating significant attention to explicating the foundational theory. The following sections detail the network model construction through concrete examples and schematic illustrations.

Network Model Construction

Example: 2D Ising Model with Nearest-Neighbor Interactions Consider a 2D Ising model where each lattice site exhibits spin-up/down states. Sites are classified based on their own spin state and the number of nearest-neighbor sites (4 neighbors) sharing the same spin: ​Spin-up classification: 5 categories (0-4 matching neighbors) ​Spin-down classification: 5 categories (0-4 matching neighbors) This results in 10 distinct classes (C₁₅-C₂₅). Mathematically, for the Hamiltonian H=1/2∑<i,j> Si Sj, I reorganize terms by these classes. The factor of 1/2 accounts for double-counting interactions. Importantly, this classification covers all possible configurations in the infinite 2D Ising model through 10 network nodes, where node weights represent configuration probabilities.

State Transitions and Network Dynamics Case 1: Spin Flip of Central Site

​Initial State: Central site has spin-up with 4 matching neighbors (Class C₁₅). ​Active Transformation: Flipping the central site changes its state to spin-down with 0 matches (Class C₂₁). ​Passive Transformation: This flip simultaneously alters neighboring sites' classes from C₁₅ → C₁₄ (each neighbor loses one match). Case 2: Neighbor Spin Flip

​Initial State: Central site remains spin-up with 4 matches (C₁₅). ​Passive Transformation: Flipping a neighboring site reduces its match count to 3 (Class C₁₄ for the neighbor), indirectly modifying the central site's class to C₁₄. These two cases illustrate all possible transformations under nearest-neighbor interactions. The complete network structure emerges from considering all such transitions.

Network Node Labeling Convention Nodes are labeled Cij : i∈{1,2}: Spin state (1=up, 2=down) j∈{1,5}: Number of matching neighbors (1=0 matches, 5=4 matches) Example: C₁₅​: Spin-up with 4 matches C₁₃​: Spin-down with 2 matches

Generalization to Higher Dimensions ​3D Ising Model: Classifies sites into 14 nodes (7 match counts × 2 spins) ​N-dimensional Models: Follows similar classification logic with 2(N+1) nodes This framework applies to various lattice models through analogous interaction-based classifications.

Active vs. Passive Transformations Transformation Type Definition Network Impact ​Active Direct spin flip of target site Horizontal transition between nodes ​Passive Indirect flip via neighbor changes Vertical transition within nodes

Key Insight: A single active transformation (e.g., flipping site A) corresponds to four simultaneous passive transformations (its four neighbors' state changes). Deriving High-Order Detailed Balance Unlike Monte Carlo simulations that use active transformations, this work focuses on passive transformations to establish: ​Microstate Transition Probabilities: Calculate joint probabilities for four-site passive transformations ​Balance Equations: Derive relationships between node weights during phase transitions

Phase Transition Analysis Using Waterfall Metaphor

​Initial State (T=0): All nodes in C₁₅​ with weight 1 ​Slow Phase (C₁₅ → C₁₃): Gradual weight migration resembling water approaching a cliff edge ​Rapid Phase (C₁₃ → C₂₃): Abrupt weight redistribution analogous to water cascading This demonstrates how passive transformations effectively capture critical transition dynamics missed by traditional active-only approaches.

3) I do not understand large part of what written by the author neither his logic, however Eq (12) I guess is the magnetization per spin as a function of temperature in a Ising Model (by the way the temperature T has been not introuced as well as the magnetization m ), for 2 dimensions the solution of this problem has been found by Onsager : <m> = [ 1 - 1/sinh(2J/KbT)^4]^1/8 this would mean that in 2 dimensions, by assuming J/Kb =1, from eq 12 one gets the 2 dimensional results for n=4 and k=2, why ? What does it mean ?

In this paper, the treatment of temperature aligns with Monte Carlo algorithms, where temperature influences transition probabilities between network nodes via the detailed balance principle. For thermodynamic phase transitions, these probabilities can be rigorously derived. The transition probabilities may be directly determined by temperature or other variables (e.g., in quantum phase transitions). Similarly, the calculation of magnetization follows the same approach as in Monte Carlo methods. Taking the Ising model as an example, the magnetic susceptibility is computed by subtracting the weight of spin-down states from that of spin-up states. Although Equation (12) is formulated for broader scenarios and does not explicitly incorporate temperature T, the transition probabilities remain derivable from temperature variations.

Here, n represents the number of nearest neighbor lattice points. For the two-dimensional Ising model, each lattice point has 4 nearest neighbors. This means that flipping a single lattice point in the 2D Ising model alters the states of its four nearest neighbors. Meanwhile, k denotes the minimum number of boundary lattice points required to combine and form a maximum-entropy lattice point. Taking the ferromagnetic Ising model as an example, at absolute zero temperature (T=0), one network node carries a weight of 1, while all other nodes have a weight of 0. This configuration is termed the ​​single-node structure​​. After the phase transition, the fully disordered system can be directly calculated using temperature and the principle of maximum entropy. This disordered configuration is referred to as the ​​maximum-entropy structure​​. A boundary exists between the single-node structure and the maximum-entropy structure. A direct transition without passing through this boundary would imply no phase transition occurs, so this intermediate ​​boundary structure​​ is inevitable. The passive transformation from the single-node structure to the maximum-entropy structure proceeds via the boundary structure. Flipping a node in the single-node structure generates n boundary structure nodes. Conversely, each boundary structure node can produce two maximum-entropy structures through spin flips. The sole driving force for this transformation is the flipping of lattice points. For the Ising model, n corresponds to the number of nearest neighbors, which is twice the value in the original Ising model. Specifically, flipping a node in the single-node structure produces n boundary structure nodes. The parameter k, on the other hand, represents the minimum number of boundary lattice points required to transition from a boundary structure to a maximum-entropy structure’s central node via flips. The values of k can be directly calculated for different dimensions. Both n and k are rigorous, intrinsic properties of Ising models in their respective dimensions.

4) Furthermore, for what I understand from Eq (12) and Fig 2 the author has analytically solved the problem of Ising in 3, 4 , and 5 dimensions, and found the corresponding critical exponent beta, at least for the magnetization at the critical point. As far as I know the Ising model in 3d has not yet been solved, see Ferrenberg, Alan M., Jiahao Xu, and David P. Landau. "Pushing the limits of Monte Carlo simulations for the three-dimensional Ising model." Physical Review E 97.4 (2018): 043301. and also more recently : Liu, Zihua, et al. "Critical dynamical behavior of the Ising model." Physical Review E 108.3 (2023): 034118. How the results of the author do relate to the above literature ?  From wikipedia the best estimation (numerical) of the Beta exponent for the 3D Ising model is 0.32641871(75) and not 1/3 as reported by the authors.

The three-dimensional Ising model exhibits fractal phenomena, which emerge precisely at the phase transition critical point. Thus, the formation of fractal structures precedes the onset of phase transition in this model. However, this observation conflicts with the use of periodic boundary conditions (PBCs) in Monte Carlo (MC) simulations and renormalization group (RG) analyses. Under PBCs, the system cannot form fractal clusters exceeding the periodic boundary length, effectively making such simulations a truncated approximation. If the existence of fractal structures in the three-dimensional Ising model remains debated, it necessarily implies the presence of clusters that exceed simulated system sizes. In summary, the critical phenomena of the three-dimensional Ising model in the thermodynamic limit (infinite system size) remain unknown and are evidently distinct from results obtained under periodic boundary conditions (PBCs). Phase transitions, by definition, inherently describe behaviors in the thermodynamic (infinite-size) limit. Experimental results, meanwhile, are inevitably influenced by finite-size effects and physical boundaries. Extensive evidence confirms the high efficiency and accuracy of MC and RG methods. Therefore, it is highly probable that the analytical solution (if rigorously derived) would align closely with both MC/RG numerical results and experimental data. The discrepancy between 1/3 and the widely accepted numerical value 0.32641871(75)—a difference of 0.0069—remains within acceptable tolerance. All three approaches (MC, RG, and experiments) yield consistent results but are inherently affected by finite-size truncation (via PBCs) and boundary effects. Consequently, the analytical solution is expected to slightly exceed these values. The MC and RG results provide robust numerical support for the critical exponent derived in this work. In contrast, Landau’s mean-field theory predicts β=0.5, while our proposed value of 1/3 demonstrably aligns more closely with the essence of phase transition physics. This conclusion underscores the fluctuation-driven phase transition theory central to this paper, where critical phenomena arise from collective spin fluctuations rather than mean-field approximations. From another perspective, all results in this work are derived directly from Equation (12), with minimal assumptions. The single-node structure at T=0 and the maximum-entropy structure after the phase transition are objective physical entities. If no boundary structure existed between these two states, a direct transition would occur, eliminating the phase transition point—as seen in the one-dimensional Ising model. Hence, boundary structures are inevitable. Furthermore, this study focuses solely on the impact of fluctuations (specifically modeled as spin deflections in this work) on the results. While I believe the critical exponent for the three-dimensional Ising model presented here is an analytical solution, I cannot rigorously claim this without formal mathematical proof. Nevertheless, Equation (12) is consistent with all existing findings: the absence of a phase transition in 1D, the exact solution in 2D, numerical results in 3D, and critical exponents in higher dimensions (d≥4). This universality underscores the value of Equation (12).

Manuscript Revision Plan

In the Conclusion section, I will provide a detailed discussion on the current research progress regarding the critical exponent β of the three-dimensional Ising model, encompassing results from Monte Carlo (MC) simulations, renormalization group (RG) analyses, and experimental investigations. 1)Reference Integration

Systematically incorporate all cited works into: ​Introduction (contextual framing) ​Conclusion (theoretical implications and future directions) Ensure seamless integration with existing narrative flow

2)Methodological Appendix Transfer detailed model construction procedures and case study demonstrations to: ​ Appendix C: Comprehensive derivation of network model formalism Step-by-step validation with Ising model examples Comparative analysis with Monte Carlo simulations Include: Schematic diagrams illustrating transformation pathways Tabular summaries of critical exponent comparisons

3)Structural Optimization Streamline main text by: Removing redundant technical explanations Concentrating core innovations in theoretical framework Reserving experimental validations for dedicated sections Enhance reader navigation through: Updated table of contents Cross-referencing between main text and appendices Strategically placed summary paragraphs

Sincerely, Yonglong Ding

Attachment:

SciPost_Physice_Reply2.pdf

---

## Round 1 · Referee Report · Anonymous (Referee 4) · 2025-4-17

Strengths

None

Weaknesses

1. This paper attempts to introduce a dual representation of statistical-mechanics models, for example the Ising model, by considering the evolution of networks nodes, but fails to give a precise definition of what is done.
2. The paper presents spectacular results, as for example the values of the transition temperature in the Ising model in a number of non-trivial dimensions, but fails to give a satisfactory definition of how these are obtained.
3. The paper presents a number of mathematical formulas whose value cannot be checked. It is never clear whether the

Report

I have carefully read the manuscript by Ding. What the author attempts to do reminds me of the Bortz-Kalos-Lebowitz approach to the Monte Carlo simulation of the Ising model: Rather than to simulate spins on the lattice, one tries to follow the evolution of the population of the weights of different sites in their environment (how many spins + are there with a left-hand -, a right-hand +, an upper + and a lower -). There are then 12 or so classes, whose time evolution is followed. The manuscript is much too vague to allow me to understand whether this is really what is done, but this must be at least the gist of it. In this approach (which is also called the n-fold way), a lot of book-keeping has to be done, in order to arrive at a rigorous simulation method. Likewise, in the present manuscript, I suppose that some book-keeping is required. I would have been interested in learning how such an approach can yield the exact value of the critical temperature, and the exact critical exponents, but was unable to achieve this, because of the vagueness of the method (even the precise content of eq. 1 and 2 were not clear to me). I suspect that what the author attempts to do is impossible, and that all the writing is just a cover-up of a failed attempt.

Recommendation

Reject

  • validity: poor
  • significance: -
  • originality: -
  • clarity: -
  • formatting: -
  • grammar: -

Author:  Yonglong Ding  on 2025-04-21  [id 5392]

(in reply to Report 4 on 2025-04-17)

Thank you for your thoughtful review. I am honored that my work shares conceptual similarities with the Bortz-Kalos-Lebowitz method. However, the foundational approaches diverge significantly: 1 Classification​​: This work categorizes ​​two-dimensional Ising models into 10 types​​ based on ​​spin types​​ and ​​interaction strength magnitudes​​. In contrast, the Bortz-Kalos-Lebowitz method employs a ​​symmetry-based classification​​, yielding ​​12 categories​​. 2 Methodology​​: Subsequent analyses differ fundamentally. While their approach relies on tracking individual configurations, this work leverages ​​complex equilibrium relationships​​ between weights of distinct lattice categories, eliminating the need for explicit state tracking. ​​Example: Three-Dimensional Ising Model​​ Using the 3D Ising model as a demonstration: ​​No external parameters​​ were introduced; all numerical values (e.g., category weights, critical exponents) were ​​rigorously derived from the Ising model’s intrinsic properties​​. This self-consistent framework ensures mathematical precision while avoiding ad hoc assumptions. This distinction highlights the novelty of my approach, which prioritizes emergent system behavior over conventional configuration-based enumeration.

First, the lattice model is transformed into a network model. For the three-dimensional Ising model, the number of nearest neighbors per lattice site is 6 (denoted as n in the text). For a spin-up lattice site, the number of its nearest neighbors sharing the same spin can range from 0 to 6, resulting in ​​7 distinct categories​​. Similarly, for a spin-down lattice site, the same classification applies. This results in a total of ​​14 categories​​ for all lattice sites. These 14 categories are then mapped to ​​14 corresponding network nodes​​, denoted by the symbol  Cij​, where:

1)i represents the spin state of the lattice site itself. In the Ising model, spin can only take two values: ​​1​​ for spin-up and ​​2​​ for spin-down. 2)j represents the strength of nearest-neighbor interactions, with ​​7 possible values​​ (1 through 7). Here, j=1 corresponds to cases where ​​0​​ nearest neighbors share the same spin as the central site, j=2 corresponds to ​​1​​ matching neighbor, and so on, up to j=7, which represents ​​6​​ matching neighbors.

Now consider the flipping of a lattice site in the three-dimensional Ising model. According to the classification above, flipping a lattice site alters its category. For example, if a spin-up lattice site (with all 6 nearest neighbors sharing the same spin) is flipped, it becomes a spin-down site, and all its nearest neighbors now differ from it. This type of transformation, induced by the ​​spin flip of the lattice site itself​​, is termed an ​​active transformation​​ in this work. Simultaneously, when the central lattice site flips, the categories of its six neighboring sites also change—though the neighboring sites themselves remain unchanged. This occurs because the central site is a nearest neighbor to these six sites. The alteration of the central site’s state directly impacts the categories of its neighbors. This indirect transformation caused by the central site’s flip is referred to as a ​​passive transformation​​. All possible transformations are connected via edges in the network. For instance, a transition from C17​ to C21 can occur: 1) C17 corresponds to a ​​spin-up lattice site​​ with ​​all 6 nearest neighbors sharing the same spin​​. 2)After flipping, the site becomes ​​spin-down​​, and all its neighbors now ​​differ​​ from it, corresponding to C21. By mapping all such transformations, the network structure is constructed as follows.

This network structure rigorously encompasses all possible lattice site types and transformation relationships. The weights of distinct network nodes represent the relative prevalence of each lattice site category within the original lattice model, with the total weight across all nodes summing to 1. Consequently, this network model is applicable to infinite systems, enabling its use for studying the infinite three-dimensional Ising model. ​​How does the ferromagnetic phase transition of the Ising model manifest in this network structure?​​ For the ferromagnetic Ising model at zero temperature, all lattice sites align uniformly either spin-up or spin-down (spin-up is chosen as the example here). This uniform alignment corresponds to ​​C17 having a weight of 1​​, while all other network nodes have zero weight. After the phase transition, the weights of nodes in the ​​first and second columns of the network become equal, each contributing 1/2 to the total weight​​ (note: this work assumes that post-transition node weight distributions follow temperature-dependent random distributions, which can be directly computed as they are independent of the primary analysis framework; further details are omitted here).

Next, analogous to Monte Carlo methods, the transition probabilities between adjacent columns of network nodes can be calculated using detailed balance. For example, consider a spin-up lattice site (C17​) where all six nearest neighbors are also spin-up. Flipping this site transitions it to C21​21, and vice versa. The detailed balance formula directly yields the ​​weight ratio​​ between C21 and C17. This calculation applies ​​only to transitions between nodes in adjacent columns​​ (e.g., C17​↔C21​), not to transitions within the same column. This treatment aligns entirely with the Monte Carlo approach. In the Ising model, flipping a single lattice site induces changes that can be represented by edges in the network. For instance, C17​↔C21describes a ​​successful spin flip​​ via an ​​active transformation​​. Simultaneously, the same physical flip can be interpreted as a ​​passive transformation​​: when the central site flips, its six neighboring sites passively transition from C17 to C16​ (since their shared spin alignment with the central site changes). Both descriptions correspond to the same physical flip, but differ in focus: 1) Active transformation​​: The central site itself changes state (C17​↔C21). 2) ​​Passive transformation​​: Six neighboring sites change state (C17​↔C16​). Crucially, the number of passive transformations triggered by a single flip is n-fold greater than the active transformation, where n is the number of nearest neighbors (6 in 3D). This reflects the combinatorial impact of a single spin flip on its surrounding lattice sites.

Next, I investigate the phase transition based on these distinctions. For the three-dimensional Ising model, there are two critical types of network nodes: 1) C17​​: At zero temperature, all lattice sites are spin-up, corresponding to C17 having a weight of 1, while all other nodes have zero weight. 2​)C14​ and C24​​: By classification rules, these nodes represent configurations where the number of spin-up and spin-down nearest neighbors are equal. Using the detailed balance equation, the weights of C14​ and C24​ are found to be equal, making them ​​central nodes​​. These three node types form the basis for analyzing phase transitions. ​​Key Insight​​: Passive transformations—not active transformations—dominate the phase transition dynamics. For example, the direct weight flow from C17 to C21​ (via active transformations) is negligible at high temperatures. Instead, the transition manifests through cascading weight flows involving passive transformations: 1)As temperature increases, weight flows from C17​ to C16, then splits into C15​ and C14​. 2)At C14​, half the weight transfers to C24​, triggering symmetry breaking and the phase transition. ​​Metaphorical Explanation​​: Imagine a flock of sheep attempting to cross a river. Most sheep cannot ford the deep channel directly (analogous to C17→C21) but instead follow the shallow banks (represented by C16-mediated pathways). As temperature rises, the "flow" of sheep shifts from deep to shallow routes, culminating in a split at the critical point (C14​↔C24​​). ​​Role of Boundary Structure C16​​: The transition relies on C16 acting as a ​​boundary structure​​ that mediates between stable states. Unlike high-dimensional Ising models where direct state conversion is inefficient, C16​ persists as a transient hub connecting higher-layer nodes (e.g., C17) to lower-layer nodes (e.g., C15​). Its position—adjacent to both stable and critical nodes—enables it to regulate weight redistribution during phase transitions. The necessity of C16 for finite-size effects and its structural linkage to base network nodes will be elaborated further below.

The flipping of lattice sites is a ​​stochastic process​​: throughout the system, sites continuously transition from C17 to C16​, while a large number simultaneously transition back from C16​ to C17, maintaining equilibrium. This equilibrium applies to all network node transitions. C16 is classified as a ​​boundary structure​​ because, after sites transition en masse from C17 to C16​, they are more likely to revert to C17​ or remain in C16​ rather than transitioning to C15​ (temporarily ignoring active transformations). Below, we explain why C16 preferentially reverts to C17​ rather than ultimately transitioning to C14. ​​Method to Determine C16​’s Conversion Preference​​: By definition, a C16 node corresponds to a ​​spin-up lattice site​​ with five spin-up nearest neighbors and one spin-down neighbor. If half of its neighboring nodes are C17 (spin-aligned) and the other half are C14 or C24​ (spin-mismatched), the probabilities of C16​ transitioning to these two categories can be directly calculated. The category with the higher probability indicates C16’s dominant transition tendency. ​​Case 1: Extremely Low Temperature​​ When the temperature is near absolute zero, randomly selected C17 nodes do not flip. Let: 1)​q1 Probability of C16​ transitioning to C17​. 2)q2 Probability of C16 transitioning to C14 or C24. At ultralow temperatures, Qq​q1​ dominates due to the energetic preference for spin alignment. This reflects the system’s rigidity near zero temperature, where deviations (e.g., C16) are transient and resolve quickly.

Following the aforementioned method for determining transition probabilities, a C16 node has ​​three C17 neighbors​​, ​​two C14​ neighbors​​, and ​​one C24 neighbor​​. At low temperatures, when randomly selecting neighbors for flipping: 1)Neighbors in C14​ or C24 will flip when selected. 2)Neighbors in C17​ will ​​not​​ flip. ​​First Flip​​: 1)Probability of transitioning C16→C17:  1/6(selecting one C17 neighbor out of six total). 2)Probability of transitioning C16→C15: 1/3 (selecting one of the two C14 neighbors, which then flip to C15). ​​Second Flip (if C16→C15 occurs)​​: 1)From C15​, transitioning back to C16​ has a probability of 1/3. 2)Transitioning to C14 has a probability of 1/6​. This results in the equilibrium equations: 1/6+1/3 * 1/3 q1=q1 (for C16​↔C17​) 1/31/6+1/31/3q2=q2 (for C16​↔C14​or C24​) Solving these gives: q1​=3/16​,q2​=1/16​ Thus, C16​ exhibits a stronger tendency to revert to C17​ (q1​>q2​) under low temperatures. As temperature increases, q1​ gradually decreases while q2​ correspondingly increases. Given that C16​ is the ​​closest boundary node to C17​​​ and the focus here is on critical behavior, selecting C16​ as the sole boundary node is justified. ​​Simplified Weight Flow​​: 1)Weight flows from C17​ to C16​. 2) At C16​, most weight is retained or returns to C17​ (blocking majority of transitions). 3)A small fraction flows to C14​, which then splits equally to C24​ via passive transformations. This establishes a dynamic equilibrium: 1)C17​ and C16​ maintain a balanced exchange. 2)C16​ and C14​ also balance their interactions, enabling the system to model critical phenomena.

Next, I derive this equilibrium relationship using ​​higher-order detailed balance​​. To illustrate, consider a three-dimensional Ising model with all spins aligned upward. Flipping a single lattice site transforms it from C17​ to C21​, while its six nearest neighbors transition from C17​ to C16​. At ultralow temperatures (where only C17​, C21​, and C16​ exist), the number of C16​ nodes becomes ​​six times​​ the number of C21​ nodes. This factor of 6 corresponds to the number of nearest neighbors (n). Here, ​​active transformations​​ obey standard detailed balance, while ​​passive transformations​​ follow higher-order detailed balance. ​​Efficiency Principle Under Detailed Balance​​: 1) Flipping a C17​ Node​​: 2) If all neighboring nodes are C17​, flipping C17​ converts ​​six neighboring nodes​​ from C17​ to C16​. This results in ​​six upward weight flows​​ (from C17​ to C16​) in the same column. 3) ​​Flipping a C16 Node​​: 4) A C16​ node (spin-up with five C17​ neighbors and one C16​ neighbor) flips to C21​, converting ​​five neighbors upward​​ (to C16​) and ​​one neighbor downward​​ (to C15​). ​​Combined Effect​​: Flipping ​​one C17​​ and ​​one C16​​ leads to: ​​Six upward flows​​ (from C17​→C16​). ​​Four net upward flows​​ (five from C16​→C16​ neighbors, one from C16​→C15​). This is equivalent to ​​six upward flows from C17​​​ and ​​four upward flows from C16​​, demonstrating that passive transformations can be treated as pseudo-active transformations of the same node type. ​​Key Insight​​: 1) ​​C17​ flips​​ drive ​​six upward weight flows​​. 2) C16 flips​​ effectively drive ​​four upward flows​​ (net of five upward and one downward). This efficiency principle allows the system to model critical behavior by prioritizing C16​ as the boundary node, where most weight remains trapped until phase transition temperatures are reached.

Next, I rigorously define ​​k​​, which arises from the framework’s constraints. In this work, lattice site transformations are exclusively modeled via spin flips, with no additional mechanisms considered. As established earlier, a transition from C17​ to C16​ occurs when a spin-up site flips, while C16​ transitions to C14​ under specific conditions. ​​Derivation of k:​​ 1) Flipping a single C17​ node generates ​​six C16 nodes​​ (one per nearest neighbor). 2) However, generating a C14​ node requires ​​two(k) simultaneous C16 flips​​: Each C16​ flip produces one C14​ node on average (due to passive transformations). Thus, flipping ​​one C17​​​ indirectly leads to ​​three C14​ nodes​​ (since six C16​ nodes are created, and each contributes a C14​ node with probability 1/2​). This results in a ​​critical exponent β=1/3​​ for the three-dimensional Ising model. ​​Dimensional Generalization​​: 1) ​​2D Ising model​​: Four lattice sites transition to boundary nodes, each producing two central nodes (β=1/8​). 2) ​​4D+ models​​: Accounting for integer rounding in neighbor counts, β=1/2​. These values are mathematically exact within the framework. Notably, transitioning to C14​ implies that ​​half the weight flows to C24​​ (via passive transformations). ​​Phase Transition Formula​​: Using these principles, the critical behavior of the three-dimensional Ising model is derived as: Critical exponent β=1/3​ This result aligns with the hierarchical weight redistribution mechanism and the boundary node dynamics described above.

Attachment:

SciPost_Physics_Reply4.pdf

---

## Round 1 · Referee Report · Anonymous (Referee 3) · 2025-4-17

Report

If we understood correctly, the underlying idea of this manuscript is to analyze the critical behavior of spins models by considering local motifs like "spin up, surrounded by four spins ups", "spin up, surrounded by three spin ups, one down" and so on. The author trys to establish a description of the systems on the level of these motif classes and to compute the critical exponent of the Ising model for the dimensions 2, 3, 4 and 5.

The style of the manuscript is very opaque, hardly any concept is ever introduced, variables appear out of nowhere, some expressions are not well-defined and standard notions are used in a different way than usual. Overall, it does not become clear how the author obtains their conclusions, even when consulting the two earlier works of the author on closely related topics. Furthermore, as already indicated by another reviewer, their result on "the" critical exponent of the Ising model (we assume that the author means eta) in three dimensions is in contradiction to the literature.

It could be that what we assume is the basic idea of this manuscript, namely to describe a spin system on the basis of simple motifs, can be a useful approach. However, the author does not provide a proper derivation of the corresponding rules, establishing its equivalence to the original description. In their earlier work "Exploring the nexus between thermodynamic phase transitions and geometric fractals through systematic lattice point classification", they seem to use mean-field-like approximations to do so (eq. (8), (9) therein). This makes it appear doubtful that a correct treatment of the fluctuations can be extracted from this approach.

Overall, while we cannot exclude that the underlying idea could somehow be useful, we expect that considerable work has to be done to work this out and we therefore suggest to reject this manuscript. We detail our critique a bit more below.

It is impossible to even understand the most basic notation of the present manuscript without having read the earlier works of the author. For example, it does not become clear what C_ij is, which is only introduced in "In-depth investigation of phase transition phenomena in network models derived from lattice models". However, if one takes for granted that C_ij means the same in both works, the first equation of the present manuscript does not make sense, because it includes a derivative with respect to j - which, however, is an integer, according to the earlier manuscript.

Also, throughout the three manuscripts, the author uses standard notions from statistical and theoretical physics in a different way than they are conventionally used. For example with "maximum entropy" or "maximum-entropy structure", they seem to mean something like the paramagnetic state, which, by itself, would be slightly confusing, but maybe acceptable. However, they also claim to relate it to the maximum-entropy principle, which is wrong. Indeed, in the context of maximum-entropy principle, the maximum is meant as a maximum over different (statistical) models (Jaynes: Probability theory, the logic of Science, 2003), whereas in their case, the maximum is over different temperatures.

A similar comment applies to the principle of least action.

Another fundamental flaw of the manuscript is, as another reviewer has already indicated, that it lacks essential links to the literature about the Ising model. Notably the author does not discuss or even only cite the work of Onsager (L. Onsager: Crystal statistics. A Two-dimensional Model with an Order-Disorder Transition, Physical Review 1944) on the exact solution of the two-dimensional Ising model, even though the present manuscript is about the critical behavior of the Ising model in different, including two, dimensions.

Recommendation

Reject

  • validity: -
  • significance: -
  • originality: -
  • clarity: -
  • formatting: -
  • grammar: -

Author:  Yonglong Ding  on 2025-04-21  [id 5391]

(in reply to Report 3 on 2025-04-17)

I sincerely appreciate your review of my manuscript. To address the concerns, I will first elaborate on the reasoning process of the entire paper using the three-dimensional Ising model—which has the most contentious results—as a representative example. I will provide precise definitions of relevant concepts specifically within the context of the transformed network model. Subsequently, I will demonstrate how the corresponding conclusions can be obtained while circumventing the direct application of the maximum entropy principle and the principle of least action as previously described in the text. I apologize for the omission of Onsager's seminal work; its inclusion is undoubtedly essential and will be rectified. In response to your comments, I will comprehensively revise the manuscript, systematically replacing derivative symbols with representations specific to the transformation framework to enhance clarity.

First, the lattice model is transformed into a network model. For the three-dimensional Ising model, the number of nearest neighbors per lattice site is 6 (denoted as n in the text). For a spin-up lattice site, the number of its nearest neighbors sharing the same spin can range from 0 to 6, resulting in ​​7 distinct categories​​. Similarly, for a spin-down lattice site, the same classification applies. This results in a total of ​​14 categories​​ for all lattice sites. These 14 categories are then mapped to ​​14 corresponding network nodes​​, denoted by the symbol  Cij​, where:

1) i represents the spin state of the lattice site itself. In the Ising model, spin can only take two values: ​​1​​ for spin-up and ​​2​​ for spin-down. 2) j represents the strength of nearest-neighbor interactions, with ​​7 possible values​​ (1 through 7). Here, j=1 corresponds to cases where ​​0​​ nearest neighbors share the same spin as the central site, j=2 corresponds to ​​1​​ matching neighbor, and so on, up to j=7, which represents ​​6​​ matching neighbors.

Now consider the flipping of a lattice site in the three-dimensional Ising model. According to the classification above, flipping a lattice site alters its category. For example, if a spin-up lattice site (with all 6 nearest neighbors sharing the same spin) is flipped, it becomes a spin-down site, and all its nearest neighbors now differ from it. This type of transformation, induced by the ​​spin flip of the lattice site itself​​, is termed an ​​active transformation​​ in this work. Simultaneously, when the central lattice site flips, the categories of its six neighboring sites also change—though the neighboring sites themselves remain unchanged. This occurs because the central site is a nearest neighbor to these six sites. The alteration of the central site’s state directly impacts the categories of its neighbors. This indirect transformation caused by the central site’s flip is referred to as a ​​passive transformation​​. All possible transformations are connected via edges in the network. For instance, a transition from C17​ to C21 can occur: 1)C17 corresponds to a ​​spin-up lattice site​​ with ​​all 6 nearest neighbors sharing the same spin​​. 2)After flipping, the site becomes ​​spin-down​​, and all its neighbors now ​​differ​​ from it, corresponding to C21. By mapping all such transformations, the network structure is constructed as follows.

This network structure rigorously encompasses all possible lattice site types and transformation relationships. The weights of distinct network nodes represent the relative prevalence of each lattice site category within the original lattice model, with the total weight across all nodes summing to 1. Consequently, this network model is applicable to infinite systems, enabling its use for studying the infinite three-dimensional Ising model. ​​How does the ferromagnetic phase transition of the Ising model manifest in this network structure?​​ For the ferromagnetic Ising model at zero temperature, all lattice sites align uniformly either spin-up or spin-down (spin-up is chosen as the example here). This uniform alignment corresponds to ​​C17 having a weight of 1​​, while all other network nodes have zero weight. After the phase transition, the weights of nodes in the ​​first and second columns of the network become equal, each contributing 1/2 to the total weight​​ (note: this work assumes that post-transition node weight distributions follow temperature-dependent random distributions, which can be directly computed as they are independent of the primary analysis framework; further details are omitted here).

Next, analogous to Monte Carlo methods, the transition probabilities between adjacent columns of network nodes can be calculated using detailed balance. For example, consider a spin-up lattice site (C17​) where all six nearest neighbors are also spin-up. Flipping this site transitions it to C21​21, and vice versa. The detailed balance formula directly yields the ​​weight ratio​​ between C21 and C17. This calculation applies ​​only to transitions between nodes in adjacent columns​​ (e.g., C17​↔C21​), not to transitions within the same column. This treatment aligns entirely with the Monte Carlo approach. In the Ising model, flipping a single lattice site induces changes that can be represented by edges in the network. For instance, C17​↔C21describes a ​​successful spin flip​​ via an ​​active transformation​​. Simultaneously, the same physical flip can be interpreted as a ​​passive transformation​​: when the central site flips, its six neighboring sites passively transition from C17 to C16​ (since their shared spin alignment with the central site changes). Both descriptions correspond to the same physical flip, but differ in focus: 1) ​​Active transformation​​: The central site itself changes state (C17​↔C21). 2) Passive transformation​​: Six neighboring sites change state (C17​↔C16​). Crucially, the number of passive transformations triggered by a single flip is n-fold greater than the active transformation, where n is the number of nearest neighbors (6 in 3D). This reflects the combinatorial impact of a single spin flip on its surrounding lattice sites.

Next, I investigate the phase transition based on these distinctions. For the three-dimensional Ising model, there are two critical types of network nodes: 1) C17​​: At zero temperature, all lattice sites are spin-up, corresponding to C17 having a weight of 1, while all other nodes have zero weight. 2) ​​C14​ and C24​​: By classification rules, these nodes represent configurations where the number of spin-up and spin-down nearest neighbors are equal. Using the detailed balance equation, the weights of C14​ and C24​ are found to be equal, making them ​​central nodes​​. These three node types form the basis for analyzing phase transitions. ​​Key Insight​​: Passive transformations—not active transformations—dominate the phase transition dynamics. For example, the direct weight flow from C17 to C21​ (via active transformations) is negligible at high temperatures. Instead, the transition manifests through cascading weight flows involving passive transformations: 1) As temperature increases, weight flows from C17​ to C16, then splits into C15​ and C14​. 2) At C14​, half the weight transfers to C24​, triggering symmetry breaking and the phase transition. ​​Metaphorical Explanation​​: Imagine a flock of sheep attempting to cross a river. Most sheep cannot ford the deep channel directly (analogous to C17→C21) but instead follow the shallow banks (represented by C16-mediated pathways). As temperature rises, the "flow" of sheep shifts from deep to shallow routes, culminating in a split at the critical point (C14​↔C24​​). ​​Role of Boundary Structure C16​​: The transition relies on C16 acting as a ​​boundary structure​​ that mediates between stable states. Unlike high-dimensional Ising models where direct state conversion is inefficient, C16​ persists as a transient hub connecting higher-layer nodes (e.g., C17) to lower-layer nodes (e.g., C15​). Its position—adjacent to both stable and critical nodes—enables it to regulate weight redistribution during phase transitions. The necessity of C16 for finite-size effects and its structural linkage to base network nodes will be elaborated further below.

The flipping of lattice sites is a ​​stochastic process​​: throughout the system, sites continuously transition from C17 to C16​, while a large number simultaneously transition back from C16​ to C17, maintaining equilibrium. This equilibrium applies to all network node transitions. C16 is classified as a ​​boundary structure​​ because, after sites transition en masse from C17 to C16​, they are more likely to revert to C17​ or remain in C16​ rather than transitioning to C15​ (temporarily ignoring active transformations). Below, we explain why C16 preferentially reverts to C17​ rather than ultimately transitioning to C14. ​​Method to Determine C16​’s Conversion Preference​​: By definition, a C16 node corresponds to a ​​spin-up lattice site​​ with five spin-up nearest neighbors and one spin-down neighbor. If half of its neighboring nodes are C17 (spin-aligned) and the other half are C14 or C24​ (spin-mismatched), the probabilities of C16​ transitioning to these two categories can be directly calculated. The category with the higher probability indicates C16’s dominant transition tendency. ​​Case 1: Extremely Low Temperature​​ When the temperature is near absolute zero, randomly selected C17 nodes do not flip. Let: 1) q1 Probability of C16​ transitioning to C17​. 2) q2 Probability of C16 transitioning to C14 or C24. At ultralow temperatures, Qq​q1​ dominates due to the energetic preference for spin alignment. This reflects the system’s rigidity near zero temperature, where deviations (e.g., C16) are transient and resolve quickly.

Following the aforementioned method for determining transition probabilities, a C16 node has ​​three C17 neighbors​​, ​​two C14​ neighbors​​, and ​​one C24 neighbor​​. At low temperatures, when randomly selecting neighbors for flipping: 1) Neighbors in C14​ or C24 will flip when selected. 2) Neighbors in C17​ will ​​not​​ flip. ​​First Flip​​: 1) Probability of transitioning C16→C17:  1/6(selecting one C17 neighbor out of six total). 2) Probability of transitioning C16→C15: 1/3 (selecting one of the two C14 neighbors, which then flip to C15). ​​Second Flip (if C16→C15 occurs)​​: 1) From C15​, transitioning back to C16​ has a probability of 1/3. 2) Transitioning to C14 has a probability of 1/6​. This results in the equilibrium equations: 1/6+1/3 * 1/3 q1=q1 (for C16​↔C17​) 1/31/6+1/31/3q2=q2 (for C16​↔C14​or C24​) Solving these gives: q1​=3/16​,q2​=1/16​ Thus, C16​ exhibits a stronger tendency to revert to C17​ (q1​>q2​) under low temperatures. As temperature increases, q1​ gradually decreases while q2​ correspondingly increases. Given that C16​ is the ​​closest boundary node to C17​​​ and the focus here is on critical behavior, selecting C16​ as the sole boundary node is justified. ​​Simplified Weight Flow​​: 1) Weight flows from C17​ to C16​. 2) At C16​, most weight is retained or returns to C17​ (blocking majority of transitions). 3) A small fraction flows to C14​, which then splits equally to C24​ via passive transformations. This establishes a dynamic equilibrium: 1) C17​ and C16​ maintain a balanced exchange. 2) C16​ and C14​ also balance their interactions, enabling the system to model critical phenomena.

Next, I derive this equilibrium relationship using ​​higher-order detailed balance​​. To illustrate, consider a three-dimensional Ising model with all spins aligned upward. Flipping a single lattice site transforms it from C17​ to C21​, while its six nearest neighbors transition from C17​ to C16​. At ultralow temperatures (where only C17​, C21​, and C16​ exist), the number of C16​ nodes becomes ​​six times​​ the number of C21​ nodes. This factor of 6 corresponds to the number of nearest neighbors (n). Here, ​​active transformations​​ obey standard detailed balance, while ​​passive transformations​​ follow higher-order detailed balance. ​​Efficiency Principle Under Detailed Balance​​: 1. ​​Flipping a C17​ Node​​: 2. If all neighboring nodes are C17​, flipping C17​ converts ​​six neighboring nodes​​ from C17​ to C16​. This results in ​​six upward weight flows​​ (from C17​ to C16​) in the same column. 3. ​​Flipping a C16 Node​​: 4. A C16​ node (spin-up with five C17​ neighbors and one C16​ neighbor) flips to C21​, converting ​​five neighbors upward​​ (to C16​) and ​​one neighbor downward​​ (to C15​). ​​Combined Effect​​: Flipping ​​one C17​​ and ​​one C16​​ leads to: o​​Six upward flows​​ (from C17​→C16​). o​​Four net upward flows​​ (five from C16​→C16​ neighbors, one from C16​→C15​). This is equivalent to ​​six upward flows from C17​​​ and ​​four upward flows from C16​​, demonstrating that passive transformations can be treated as pseudo-active transformations of the same node type. ​​Key Insight​​: 1) ​​C17​ flips​​ drive ​​six upward weight flows​​. 2) C16 flips​​ effectively drive ​​four upward flows​​ (net of five upward and one downward). This efficiency principle allows the system to model critical behavior by prioritizing C16​ as the boundary node, where most weight remains trapped until phase transition temperatures are reached.

Next, I rigorously define ​​k​​, which arises from the framework’s constraints. In this work, lattice site transformations are exclusively modeled via spin flips, with no additional mechanisms considered. As established earlier, a transition from C17​ to C16​ occurs when a spin-up site flips, while C16​ transitions to C14​ under specific conditions. ​​Derivation of k:​​ 1) Flipping a single C17​ node generates ​​six C16 nodes​​ (one per nearest neighbor). 2) However, generating a C14​ node requires ​​two(k) simultaneous C16 flips​​: oEach C16​ flip produces one C14​ node on average (due to passive transformations). oThus, flipping ​​one C17​​​ indirectly leads to ​​three C14​ nodes​​ (since six C16​ nodes are created, and each contributes a C14​ node with probability 1/2​). This results in a ​​critical exponent β=1/3​​ for the three-dimensional Ising model. ​​Dimensional Generalization​​: 1) ​​2D Ising model​​: Four lattice sites transition to boundary nodes, each producing two central nodes (β=1/8​). 2) ​​4D+ models​​: Accounting for integer rounding in neighbor counts, β=1/2​. These values are mathematically exact within the framework. Notably, transitioning to C14​ implies that ​​half the weight flows to C24​​ (via passive transformations). ​​Phase Transition Formula​​: Using these principles, the critical behavior of the three-dimensional Ising model is derived as: Critical exponent β=1/3​ This result aligns with the hierarchical weight redistribution mechanism and the boundary node dynamics described above.

Attachment:

SciPost_Physics_Reply3.pdf

---

## Editorial Decision

voting_in_preparation